# Detection of Chicken Respiratory Pathogens in Live Markets of Addis Ababa, Ethiopia, and Epidemiological Implications

**DOI:** 10.3390/vetsci9090503

**Published:** 2022-09-14

**Authors:** Tadiose Habte Tekelemariam, Stephen Walkden-Brown, Fekadu Alemu Atire, Dessalegne Abeje Tefera, Dawit Hailu Alemayehu, Priscilla F. Gerber

**Affiliations:** 1National Poultry Research Program, Ethiopian Institute of Agricultural Research, Debrezeite P.O. Box 32, Ethiopia; 2Animal Science, School of Environmental and Rural Science, University of New England, Armidale, NSW 2351, Australia; 3Armauer Hansen Research Institute, Addis Ababa 1005, Ethiopia

**Keywords:** live market, serology, bacteriology, Sanger sequencing, mixed respiratory infection

## Abstract

**Simple Summary:**

It is unclear what microorganisms are associated with respiratory disease in chickens sold in live markets of Addis Ababa, Ethiopia. Identifying microorganisms in diseased animals is the first step to delineating programs to control disease spread. Two Addis Ababa markets were visited weekly for three months to understand better the microorganisms found in chickens with respiratory disease. During this time, 18 sick chickens were acquired and tested for common microorganisms that cause respiratory disease in chickens. Three or more microorganisms, including viruses and bacteria, were detected in 17 of the 18 cases, showing that detection of multiple pathogens is widespread in live markets. These microorganisms possibly cause substantial productive loss in Ethiopia. Further studies are warranted to investigate their contribution to disease and economic losses in the country.

**Abstract:**

A moderate to high seroprevalence of exposure to Newcastle disease (NDV), avian metapneumovirus (aMPV), infectious laryngotracheitis virus (ILTV), infectious bronchitis virus (IBV) and *Mycoplasma gallisepticum* (MG) has recently been reported in Ethiopia, but it is unclear to what extent these contribute to clinical cases of respiratory disease. This study investigated the presence of these pathogens in chickens exhibiting respiratory disease in two live markets in Addis Ababa. Markets were visited weekly for three months, and 18 chickens displaying respiratory clinical signs were acquired. Swab samples were taken from the choana, trachea, air sac and larynx for bacteriology and PCR tests targeting these five pathogens. PCR-positive samples were sequenced. All 18 chickens were PCR-positive for aMPV, 50% for each of Mg and NDV, 39% for IBV and 11% for ILTV. Infections with >3 pathogens were detected in 17 of 18 chickens. Potentially pathogenic bacteria such as *Escherichia coli, Klebsiella* spp., *Streptococcus* spp. and *Staphylococcus* were found in 16 to 44% of chickens. IBV-positive samples were of the 793B genotype. The results associate the presence of these organisms with clinical respiratory disease and are consistent with recent serological investigations, indicating a high level of exposure to multiple respiratory pathogens.

## 1. Introduction 

Poultry production in Ethiopia plays an essential role in the local economy by improving food security and alleviating poverty [1]. The majority of the poultry production in Ethiopia derives from a smallholder scavenging production system (>96%), followed by small and medium-scale intensive production [2,3]. Chickens raised in small-scale and scavenging production systems are usually sold in regional open live chicken markets, generally available only when cultural and religious festivals occur [4,5]. However, the open markets in Addis Ababa are active throughout the year and serve as a central market for adjacent regions [6,7]. There are two types of chicken markets in Addis Ababa; the first is a specialised chicken market (e.g., Shola chicken market), and the second is a general market with chickens kept in one separate section of the market (e.g., Merkato). 

Open markets provide conditions for disease transmission between birds and from birds to people, as there is often a lack of cleaning and disinfection of the facilities, equipment and personal protective garments [8,9]. Because of that, several studies have taken place in live markets to detect zoonotic pathogens such as avian influenza virus and chicken pathogens such as Newcastle disease virus (NDV) [9,10,11], Ethiopian farmers may use live chicken markets to sell apparently healthy chickens during a disease outbreak to offset losses [12], which may increase disease transmission in live markets.

Respiratory signs are commonly observed in chickens in open markets, and in particular, there is evidence of chronic respiratory diseases [13,14,15]. The aetiology of respiratory disease is complex, often involving more than one pathogen simultaneously [16]. In Ethiopia, Newcastle disease-like clinical signs are referred to as “fengil” and are characterised by dorsal prostration, sneezing, discharge from nostrils, high morbidity and mortality [8]. However, these clinical signs are non-specific and could be caused or predisposed to by several respiratory pathogens.

Previous studies on respiratory diseases of chicken in Ethiopia have shown that there was serological evidence of the high distribution of NDV in the country [17,18,19,20,21,22,23,24]. There has been serological and molecular evidence of the circulation of infectious bronchitis virus (IBV), avian metapneumovirus (aMPV) subtype B and *Mycoplasma gallysepticum* (Mg), and serological evidence of infectious laryngotracheitis virus (ILTV) circulation in the country [12,13,25,26,27,28,29]. However, the association between these and other pathogens in chickens displaying respiratory disease has not been investigated. A thermostable vaccine against NDV is administered in government-promoted chicken vaccination campaigns in Ethiopia [30]. No other chicken vaccine for respiratory pathogens is commercially available in Ethiopia [31].

Viral pathogens predispose chickens to secondary bacterial and fungal infections, although certain bacteria (e.g., *Serratia, Escherichia coli, Citrobacter, Streptococcus pyogenes, Klebsiella, Staphylococcus aureus, Enterobacter*) and fungi (e.g., *Aspergillus fumigatus and Aspergillus flavus*) can cause respiratory diseases on their own [15,32].

This study aimed to determine whether the viral respiratory pathogens detected in a recent serological survey of the zones around Addis Ababa (2012) and key bacterial respiratory pathogens are detected in clinical respiratory disease in chickens. Chickens showing respiratory disease in live markets at Addis Ababa were obtained and sampled for the presence of nucleic acids and antibodies against aMPV, ILTV, IBV, Mg and NDV. Bacterial culture of samples was also performed. We hypothesised that pathogens detected by serology and other methods in previous studies will be detected in clinical cases of respiratory disease, and that there will be a high prevalence of infection with multiple pathogens. 

## 2. Materials and Methods

### 2.1. Study Design and Sample Collection 

The research protocols used in this study were approved by the University of New England Animal Ethics Committee (approval number AEC19-047).

The study was conducted in the two major open live chicken markets of Addis Ababa, Ethiopia, namely, Merkato and Shola, between June and August of 2021. The markets were visited weekly, and chickens displaying respiratory signs (i.e., depression, dyspnoea, sneezing, coughing and nasal discharges with or without conjunctivitis) were purchased. There were more than ten chicken sellers in each open live market. Each merchant holds 30–100 chickens in baskets. Chickens showing illness were separated from healthy chickens and kept in a basket beside the baskets containing healthy chickens for sale. A maximum of two chickens with respiratory signs was selected from the “sick chicken basket” of each seller. If more than two chickens had respiratory signs, the most severely affected chickens were selected. Ten birds were acquired from Shola and eight from Merkato on five different days of visit to each market in August 2021, while there were no chickens with respiratory signs in June and July 2021.

All chickens sampled in this study were depressed and had breathing difficulty and nasal discharge. Some chickens with signs of conjunctivitis and sneezing were also selected. The acquired chickens were placed in a well-ventilated basket and transported to a location at least 2 km away from the market. Upon arrival, two swabs (FLOQ Swabs, COPAN, Brescia, Italy) were taken from the choanal cleft. Chickens were then humanely euthanised by cervical dislocation and necropsied in the open air according to the procedure described by [33]. Briefly, the carcass was wet with soapy water before dissection of the skin and muscles for exposure of internal organs. The trachea was observed for the presence of macroscopic lesions and opened with a sterile surgical blade. Two swabs were taken from the tracheal lumen; one was stored dry in the transport tube for PCR analysis, and the other was transferred to a medium containing skim milk, tryptone, glucose, and glycerol (STGG) for bacteriological culture [34]. Swab samples were also taken from lungs and air-sacs for bacteriological culture. The swabs were stored overnight at −20 °C until transport to the Armauer Hansen Research Institute (AHRI) laboratory. Samples for bacteriological analysis were stored at −20 °C, and swabs for molecular analysis were stored at −80 °C until further processing.

### 2.2. Laboratory Analysis 

Nucleic acid extraction and complementary (c)DNA synthesis. Each swab sample was placed in 600 µL of phosphate-buffered solution pH 7.4 (PBS), incubated at room temperature for 15 min and then vortexed for ten seconds. Equal volumes (200 µL) from choanal and tracheal swab washes from the same bird were pooled and homogenised, and 200 µL from the pooled sample was used for nucleic acid extraction using a commercial kit (GeneJET Viral DNA/RNA Purification Kit, ThermoFisher, Vilnius, Lithuania) according to the manufacturer’s instructions. For the detection of aMPV, NDV and IBV genome copies (GCs), cDNA synthesis was performed using a Revert Aid First Strand cDNA Synthesis Kit (ThermoFisher, Vilnius, Lithuania) according to the manufacturer’s instructions using specific reverse primers (G6 for aMPV, SX2 for IBV and NOHR for NDV (Table 1) immediately after nucleic acid extraction. The cDNA and the remaining nucleic acid extracts were stored at −80 °C until further processing.

PCR reaction and visualisation. Previously described primers were used for the detection of aMPV, ILTV, IBV, Mg and NDV (Table 1). The same master mix was used for all reactions and was comprised of 32.3 µL molecular grade water, 5 µL 10× PCR buffer, 1.5µL 50 mM MgCl_2_, 1µL 10 mM dNTP Mix, 0.2 Taq DNA polymerase (5 U/µL) and 2.5 µL 10 µM of each forward and reverse primers and 5 µL of the nucleic acid template. For nested PCR reactions, 2 µL of the first PCR reaction was used as a template. Vaccines were used as positive controls for IBV (Vaxsafe IBV Ingham strain, Bioproperties, Australia), NDV (Lasota, NVI, Ethiopia), Mg (Vaxsafe MG ts-11 strain, Bioproperties, Australia) and ILTV (Nobilis ILT Vaccine Serva strain, MSD, Australia). A synthetic gene fragment of the aMPV target sequence (IDT, Baulkham Hills, Australia) was used as a control for aMPV. The PCR amplification was carried out using an Eppendorf thermocycler (Hamburg, Germany). The amplicons were examined on a 1.7% agarose gel containing 3% ethidium bromide and visualised using a Bio-Rad Gel Doc XR+ UV Gel Documentation Imaging System (Universal Hood II). Samples were considered positive when they produced a band of the expected size (Table 1). 

Bacteriological analysis. The swab samples in STGG media were streaked into blood agar, mannitol salt agar and chocolate agar and incubated at 37 °C for 24–48 h. After culture, isolated colonies were subcultured on MacConkey agar, blood agar, Simmons citrate agar and nutrient agar for isolation of pure colonies at 37 °C for 24 h. Bacterial species were confirmed using Gram-staining and biochemical tests.

Sequencing and phylogenetic analysis. Sequencing of amplicons with the correct size for each pathogen was performed on both strands at the Ethiopian Public Health Institute, Addis Ababa. Sequences were aligned with published data using the basic local alignment search tool (BLAST) at the National Centre for Biotechnology Information (NCBI, http://www.ncbi.nlm.nih.gov/). Sequences were compiled using Geneious prime (ver.2020.0.3) software (Biomatters, Shortland Street Auckland, New Zealand) and the MEGA MUSCLE alignment algorithm. Phylogenetic analysis was performed by MEGA ver. 11 software [43].

## 3. Results 

### 3.1. Nucleic Acid Detection of Respiratory Pathogens in Respiratory Tract Swabs

Multiple pathogens were detected in all chickens sampled from the live market (Table 2). All 18 pooled swab samples were positive for aMPV type B RNA, while only two (11%) were positive for ILTV DNA. IBV, NDV and Mg nucleic acid were detected between 39 and 50% of chickens (Table 2). Chickens were positive for two (16/18, 88.9%), three (10/18, 55.6%) or four (2/18, 11.1%) of the tested pathogens by PCR.

The two ILTV-positive cases were detected in the Shola market on chickens sampled on two consecutive days. Both pooled swabs were positive for the ILTV TK gene, while only one of them was positive for the ICP4 gene. From the 9 out of 18 chickens positive for Mg, 5 chickens were positive for both C2 and 16S genes, 1 chicken was positive only for the 16S gene, and 3 additional chickens were only positive for the C2 gene. 

Sanger sequencing was performed in PCR-positive samples and confirmed specific amplification of aMPV, Mg, IBV and ILTV, while none of the NDV-positive samples could be sequenced, likely because of the low amount of genomic material. The IBV partial spike 1 gene sequences from this study (Ch 5 and Ch 10) had more than 99% homology with the MHW-Lay-Mikro-2017 isolate from Indonesia (Accession number MH671335), Quail21F1 isolate from Italy (KX077961) and isolate 37 of Greece (MG869235), which are from the 793B genotype. The current sequence had 73.9% nucleotide similarity from the DE 072 vaccinal isolate (AF274435) and GA/8077/99 strain (AF338718) from the USA, and 69.5% nucleotide similarity to the N1/88 strain isolated in Australia (U29450). 

Sample Ch 8 aMPV subtype B had more than 99.7% similarity to the aMPV isolated in Italy in 2019 (MT436233, MT436232, MT436231). The nucleotide sequences of sample Ch 10 of ILTV TK and ICP4 genes had more than 99% similarity with the Australian vaccine strain SA2 (GQ180115).

### 3.2. Culture-Based Bacteria Detection in Respiratory Tract Swabs 

Bacteria were cultured from the swabs of lungs and air sacs, showing macroscopic lesions from all but one chicken (Table 2). In total, there were 31 isolates of bacteria from 17 positive samples. Seven of 18 birds (38.9%) were positive for one bacteria species, 38.9% (7/18) were positive for two bacteria species and 16.7% (3/18) were positive for three or more bacteria species (Table 2). The most frequently isolated bacteria were *E. coli* (44.4%), *Klebsiella* (33.3%), *S. aureus* (33.3%), *Citrobacter* (27.8%), *S. pyogenes* (16.7%), *Serratia* (11.1%) and *Enterobacter* (5.5%) (Table 2). 

### 3.3. Detection of Antibodies against Respiratory Pathogens 

Eight of 18 (44%) chickens investigated were serologically negative for the five tested pathogens. Seven (39%) chickens were seropositive for aMPV and eight (44%) for IBV. Four of the IBV seropositive chickens were also IBV RNA positive. 

Three (17%) chickens were seropositive for NDV and ILTV. Of the three chickens seropositive for NDV, two were also positive by PCR. One of the ILTV PCR-positive chickens was also seropositive (Figure 1). 

Two (11%) chickens were seropositive for Mg. None of the nine chickens positive for Mg DNA was seropositive. 

## 4. Discussion

This study aimed to determine whether the viral respiratory pathogens detected in a recent serological survey of the zones around Addis Ababa [12] and bacterial respiratory pathogens are detected in clinical respiratory disease of chickens in open live markets of the same geographic area. The hypothesis was that pathogens detected by serology and other methods in previous studies will be detected in clinical cases of respiratory disease and that there will be a high prevalence of infection with multiple pathogens. The current study showed that aMPV RNA was present in all 18 chickens with respiratory disease in open markets in Addis Ababa, while 50% were positive for NDV DNA. Multiple viruses and bacteria were detected in all cases. Although there has been serological evidence of ILTV circulating in Ethiopia, this is the first study attempting to detect and then detecting the ILTV genome in the country. 

The serological results in this study provided further evidence of the circulation of aMPV, ILTV, IBV, Mg and NDV in Addis Ababa and linked their presence with clinical signs of respiratory disease, suggesting a possible role in causation. Chicken traders in Addis Ababa receive only visually healthy chickens, but it is not uncommon for chickens to exhibit signs of disease in the market before they are sold. This is not surprising given potential stress-related immunosuppression of the chickens and exposure to novel pathogens in the market. To prevent losses related to chicken disease in live markets, it is important to implement biosecurity measures and to reduce stress and overcrowding of chickens during transport to the market and while at the market. Case-control studies are required to evaluate the association of these pathogens with respiratory disease in live markets.

The chickens positive for aMPV DNA also tested positive for at least one more of the investigated pathogens. The disease caused by aMPV is known to cause immune suppression and is characterised by mild upper respiratory tract lesions and a drop in egg production, while disorientation, torticollis and opisthotonus might also be observed [44,45]. The fengil presentation characterised by respiratory signs and torticollis [8] resembles the description of severe aMPV [46]. 

The pathogens Mg and NDV were also detected using conventional PCR in 50% of the cases. Many NDV serological [17,18,19,20,21,22,23,24] and molecular investigations [47,48] have shown different NDV genotypes circulating in the rural poultry of Ethiopia. Because of the low amount of NDV DNA present in the samples in this study, it was not possible to sequence them. A previous serological investigation of Mg showed 49% seropositive chickens in East Shewa Zone [26] and 47% seropositive chickens in central Ethiopia [12]. There was also a report on the molecular presence of the MG c2 gene in sampled commercial and scavenging chickens around Bishoftu [49].

In this study, 39% of chickens showing respiratory clinical signs were positive for IBV RNA. A previous PCR study found 6% (30 of 500 chickens) positive for IBV RNA in the Jima Zone of Southwestern Ethiopia [50]. The higher prevalence in the present study (including 44% seropositive chickens) is more consistent with the high seroprevalence of IBV (> 90%) detected in a recent investigation in central Ethiopia [12]. 

Sequencing analysis showed that vaccine-like strains of ILTV and IBV were circulating in the open markets of Addis Ababa, although there are no commercially available vaccines for these pathogens available in Ethiopia. Although only two genes were sequenced from the ILTV strains, they mostly resembled the Australian ILTV SA2 vaccine strain, which is only used in Australia. The IBV sequences in this study were similar to the 793B genotype that had already been described in Ethiopia [9,42] found open live chicken markets to be a source of infection for NDV, and the current study has shown that this risk extends to other respiratory pathogens and is reflective of the presence of these pathogens in the small scale and medium scale farms of Ethiopia [12,50,51]. 

Open markets provide a mechanism for disease transmission over greater distances than the localised transmission characteristic of scavenging production systems. The combination of travel to market, the mixing of chickens and potential fomites at the market and dissemination of purchased chickens, people and fomites to new locations can facilitate the spread of new pathogens or strains. This can be exacerbated further if the stress associated with transportation and sale induces immunosuppression and if there are attempts to dispose of unhealthy chickens through the markets. Unfortunately, there are few practical options for reducing this risk without major alterations to the structure and function of the open market system in Ethiopia. 

## 5. Conclusions 

In conclusion, this study has shown that chickens with respiratory clinical signs in markets of Addis Ababa harboured multiple chicken respiratory pathogens, with the potential for dissemination of the pathogens during transport to and from the market or while housed at the market. The most commonly detected pathogen was aMPV followed by NDV and Mg, while bacterial infections were also common. For the first time, detection of the ILTV genome in Ethiopian chickens was demonstrated. The findings align with serological and other evidence of fairly widespread infection with these pathogens in Ethiopian chickens. Since there is no official vaccination program for the pathogens investigated except NDV, these diseases possibly cause a substantial productive loss in the country. Given the difficulty in controlling the transmission of such pathogens in markets, it is important to monitor the status of field-level clinical respiratory disease and causative organisms using a combination of molecular, bacteriological and serological techniques. This will facilitate practical interventions and changes in practices where warranted. 

## Figures and Tables

**Figure 1 vetsci-09-00503-f001:**
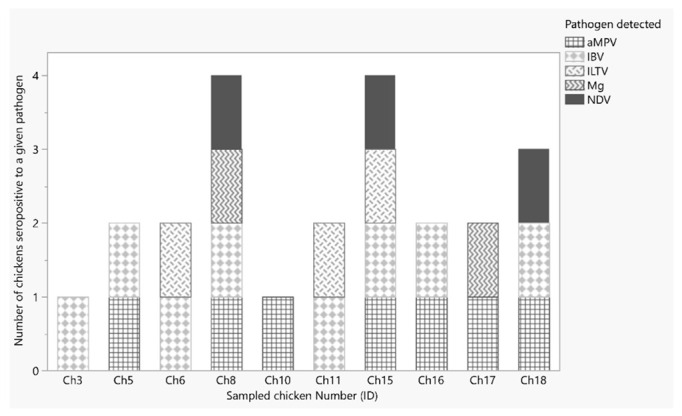
Type and number of pathogens of which chickens were seropositive.

**Table 1 vetsci-09-00503-t001:** Primers used in this study.

Pathogen	Target Gene	Primers (5′–3′)	Product Size (bp)	Reference
IBV (nested)	Spike 1 (S1) gene	PCR 1 SX1+ CACCTAGAGGTTTGT/CT A/T GCAT SX2- TCCACCTCTAAACACC C/T TT	380–393 (nested)	[35,36]
		PCR 2 SX3+ TAATACTGGC/T AATTTTTCAGA SX4- AATACAGATTGCTTACAACCAC		
aMPV (nested)	Glycoprotein (G) gene	PCR 1 G1+ GGGACAAGTATCYMAT G6- CTGACAAATTGGTCCTGATT	361 (nested)	[35,37,38]
		PCR 2 G5- CAAAGAGCCAATAAGCCCA G8+B TAGTCCTCAAGCAAGTCCTC		
NDV	Fusion (F) gene	NOHR AGT CGG AGG ATG TGT TGG CAG NOHF TAC ACC TCA TCC CAG ACA GG	260	[39]
ILTV	Thymidine kinase (TK) gene	TKF ACC TAC CTC CAA CGT ACA T TKR CCC ATA TCA GCA TTC TAG CG	395	[40,41]
	Infected cell protein 4 (ICP4)	ICP4F CTTCAGACTCCAGCTCATCTG ICP4R AGTCATGCGTCTATGGCGTTGAC	688	
Mg	16 S gene	MG-16SF GAC CTA ATC TGT AAA GTT GGT MG-16S R GCT TCC TTG CGG TTA GCA AC	186	[42]
	Cytadhesion 2 (C2) gene	Mgc2F CGC AAT TTG GTC CTA ATC CCC AAC A Mgc2R TAA ACC CAC CTC CAG CTT TAT TTC C	237–303	

**Table 2 vetsci-09-00503-t002:** Pathogen detection in swabs from respiratory organs. Each row represents molecular and bacteriological results of a pooled swab of a single chicken displaying respiratory clinical signs.

Chicken No.	Market	Date of Collection	Sex	Nucleic Acid Detection					Bacteriological Findings	Total Number of Pathogens
				MPV	IBV	ILTV	NDV	Mg		
Ch 1	Shola	09.8.21	M	+	–	–	–	+ ^x^	*S. pyogenes, Klebsiella pneumoniae*	4
Ch 2	Shola	20.8.21	M	+	–	–	+	++ ^xy^	*E. coli, Serratia*	5
Ch 3	Shola	21.8.21	M	+	+	–	+	++ ^xy^	*Citrobacter*	5
Ch 4	Shola	21.8.21	M	+	+	–	–	–	*E. coli*	3
Ch 5	Shola	23.8.21	M	+	+	–	–	–	*E. coli*	3
Ch 6	Shola	23.8.21	F	+	–	+ ^u^	+	+ ^y^	*Serratia, E. coli*	6
Ch 7	Shola	23.8.21	F	+	–	–	+	+ ^y^	*S. aureus*	4
Ch 8	Shola	24.8.21	F	+	–	–	+	–	*Citrobacter, S. aureus*	4
Ch 9	Shola	24.8.21	M	+	–	–	–	–	*Klebsiella, S. aureus*	3
Ch 10	Shola	24.8.21	F	+	+	++ ^uv^	–	–	*E. coli, S. aureus, Enterobacter*	6
Ch 11	Merkato	09.8.21	M	+	+	–	–	++ ^xy^	*S. pyogenes, Klebsiella*	5
Ch 12	Merkato	21.8.21	M	+	–	–	+	–	*Klebsiella*	3
Ch 13	Merkato	21.8.21	M	+	–	–	+	++ ^xy^	*Klebsiella*	4
Ch 14	Merkato	23.8.21	F	+	–	–	+	+ ^y^	*Citrobacter, S. aureus*	5
Ch 15	Merkato	25.8.21	F	+	–	–	+	–	-	2
Ch 16	Merkato	25.8.21	M	+	+	–	–	++ ^xy^	*E. coli*	4
Ch 17	Merkato	26.8.21	M	+	+	–	–	–	*E. coli, Citrobacter, S. pyogenes, S. aureus*	6
Ch 18	Merkato	26.8.21	F	+	–	–	–	–	*E. coli, Citrobacter, Klebsiella*	4
	Total			18	7	2	9	9		

u = ILTV gene TK; v= ILTV gene ICP; x = Mg 16S; y= Mg c2. += positive for single gene. ++=positive for two genes.

## Data Availability

The data presented in this study are available upon request from the corresponding author.

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
