# Peer review of "Detection of Chicken Respiratory Pathogens in Live Markets of Addis Ababa, Ethiopia, and Epidemiological Implications"

_vetsci, 2022, doi:10.3390/vetsci9090503_

Round 1
Reviewer 1 Report
The author investigated the presence of five pathogens in chickens exhibiting respiratory disease in two live markets in Adis Ababa. Muti-infection occured in 17/18 chickens. Bacteria were detected from lungs and air-sacs swabs of 16 chhickens. Based on nucleotide and antibody test results, aMPV and IBV are the major pathogens of live chickens in market with respiratory disease, and might originated from poultry farms. The study was able to broadly distinguish between infections and sources of respiratory pathogens of live poultry in market. The context is very clear, and the manuscript is well organized.
My comments to the authors are as follows (minor points):
1. Avian influeza A virus is a major respiratory pathogen, which was not detected in the study.
2. In the background, the immunization status of IBV and NDV in local chicken flocks should be described.
3. As 100% aMPV-positive in 18 live chickens with respiratory symptoms, It is suggested to collect throat swabs of apparently healthy chickens for aMPV detection.
Reviewer 2 Report
The authors have conducted a study to determine if the viral loads reported elsewhere (Tadiose et al., 2022) and the bacterial pathogens reported in the manuscript are associated with clinical respiratory diseases in chickens in live markets in Adis Ababa, Ethiopia.
Comment 1: The topic is interesting and relevant for the Ethiopian poultry industry and public health. Indeed, the stated presence of clinically sick birds in live markets represents a major public health issue. However, the originality of the topic for a scientific paper seems to be limited. Viral and bacterial loads will be found in clinically sick birds. Also, to support the relevance of this issue for public health and the industry, it would have been enough to survey to identify the prevalence of clinically ill birds in live markets. If no clinically ill birds were found in these markets, it would have been convenient to determine the viral and bacterial load to characterize non-obvious risks. Authors are encouraged to define their hypothesis better.
Comment 2: Please update the reference for Tadiose et al., 2022, as it is already available.
Comment 3: Please ensure all manuscript is written in the past tense.
Comment 4: Correct line 199 (Discussion)
Comment 5: Ensure that the objective at the end of the Introduction, and the aim of the study on lines 200-201 match.
Comment 6: Authors should include in Materials and Methods the methodology and criteria applied to achieve the stated objective: to determine the association between pathogens and clinically affected birds.
Comment 7: If available, present data about the prevalence of clinically affected birds in the markets.
Comment 8: when stating in lines 207-208 and 234-236 that this is the first time those pathogens have been detected, please clarify if no previous studies were conducted or if previous were negative.
Comment 9: Check the syntaxis in lines 252-253.
Comment 10: Radiation? Please clarify.
